# OpenReview forum: "Fit the Distribution: Cross-Image/Prompt Adversarial Attacks on Multimodal Large Language Models"
_NeurIPS.cc/2025/Conference — NeurIPS 2025 poster_

### Official Review · Reviewer_RVog · 2025-06-19

**Clarity:** 3
**Significance:** 3
**Originality:** 3
**Rating:** 4
**Confidence:** 4

**Summary:**

This paper proposes a generalizable cross-image/prompt adversarial attack method   from a novel distribution approximation perspective against Multimodal Large Language Models. This attack method first utilizes the Laplace approximation to model the distribution over prompts and images as two Gaussian distributions. It then employs the Monte Carlo mechanism to sample over the image and prompt distributions to optimize the image/prompt agnostic perturbation. Extensive experiments on different MLLMs and datasets validate its superior effectiveness and transferability compared with other attack methods.

**Questions:**

1.	The compared MLLMs are relatively old. Is the proposed attack method still effective against more latest models such as DeepSeekVL or DeepSeekVL-2?
2.	The paper claims that it uses all-MiniLM-L6-v2 model to compute the cosine similarity between embeddings of adversarial output and target text. Why choose all-MiniLM-L6-v2 instead of other models? Do other different models can get the same results as mentioned in the paper?
3.	The datasets used in experiments are limited. Is the proposed attack method still effective on other different datasets and other vision-language tasks?
4.	The paper uses Gaussian distributions to approximate the prompt and image distributions. Do other different data distributions have similar results?

**Ethical Concerns:**

["NO or VERY MINOR ethics concerns only"]

**Final Justification:**

After carefully reading all the responses from the authors, I have decided to keep my original score.

**Limitations:**

I suggest authors to add discussions on limitations in the paper instead of appendix.

**Quality:**

3

**Strengths And Weaknesses:**

Strengths:
1.	The paper is well-organized and easy to follow and the findings of this paper is interesting.
2.	The paper provides very detailed mathematical explanation of the procedures of the proposed attack method, thus has good mathematical theoretical support. It offers a novel method for generating effective and transferable adversarial examples against different MLLMs.
3.	The paper provides extensive and detailed experiments results to verify the effectiveness and transferability of the proposed attack method. On different MLLMs and datasets, this attack method can achieve much better cross-sample performance compared with other adversarial attack methods.
Weakness:
1.	The compared MLLMs are relatively old. The authors should test the effectiveness of the proposed attack method against more latest models such as DeepSeekVL or DeepSeekVL-2. Besides, the number of the tested MLLMs is too small, the authors should increase the number of the tested MLLMs.
2.	The paper mentions that it uses all-MiniLM-L6-v2 model to compute the cosine similarity between embeddings of adversarial output and target text. However, it does not explain the reason for using all-MiniLM-L6-v2. Does the cosine similarity computed by other models is consistent with all-MiniLM-L6-v2?
3.	The datasets used in experiments are limited. More different types of datasets are needed to further make the experiment results more convincing.
4.	The paper only compare Gaussian distribution and fixed distribution to prove the superior performance of Gaussian distribution. More different types of distributions should be considered and discussed.

---

> ### Author Rebuttal · Authors · 2025-07-30
>
> **Q1: The compared MLLMs are relatively old. The authors should test the effectiveness of the proposed attack method against more latest models such as DeepSeekVL or DeepSeekVL-2. Besides, the number of the tested MLLMs is too small, the authors should increase the number of the tested MLLMs.**
>
> **A1:** Thanks for your comment. We first want to claim that we follow the existing attacks PGD (CVPR2024), CroPA (ICLR2024), UniAtt (Neurips2024) to utilize the same MLLMs for fair comparison. Then, following your suggestion, we implement DeepSeekVL for comparison. We also implement the latest models Qwen2-VL and InternVL to provide a more comprehensive evaluation. All experiments are conducted on No Cross (N), Cross Images (I), Cross Prompts (P), and Cross Images/Prompts (IP) settings. As shown in the table below, our attack still achieves better cross-image/prompt performance than other attacks against these latest MLLM models, demonstrating our effectiveness.
>
> |Model|Method|(N) EM-ASR|(N) CM-ASR|(N) Similarity|(I) EM-ASR|(I) CM-ASR|(I) Similarity|(P) EM-ASR|(P) CM-ASR|(P) Similarity|(IP) EM-ASR|(IP) CM-ASR|(IP) Similarity|
> |:----:|:----:|:----:|:----:|:----:|:----:|:----:|:----:|:----:|:----:|:----:|:----:|:----:|:----:|
> |Qwen2-VL|PGD|63.3%|69.1%|0.676|0.0%|0.0%|0.046|28.4%|31.5%|0.334|0.0%|0.0%|0.059|
> |Qwen2-VL|CroPA|73.8%|76.6%|0.771|0.0%|0.2%|0.044|65.9%|67.5%|0.675|0.0%|0.0%|0.076|
> |Qwen2-VL|UniAtt|76.2%|79.7%|0.792|23.1%|25.6%|0.265|68.3%|69.6%|0.699|20.1%|21.5%|0.231|
> |Qwen2-VL|**Ours**|**81.2%**|**85.2%**|**0.838**|**44.6%**|**60.6%**|**0.487**|**77.2%**|**78.6%**|**0.775**|**42.7%**|**43.6%**|**0.457**|
> |InternVL|PGD|61.0%|67.4%|0.661|0.0%|0.0%|0.075|25.3%|26.8%|0.271|0.0%|0.0%|0.047|
> |InternVL|CroPA|73.1%|75.4%|0.742|0.0%|0.0%|0.086|64.6%|66.2%|0.668|0.0%|0.0%|0.081|
> |InternVL|UniAtt|75.4%|78.2%|**0.779**|22.7%|24.6%|0.248|64.8%|67.1%|0.692|21.6%|23.7%|0.245|
> |InternVL|**Ours**|**75.8%**|**79.0%**|0.745|**43.2%**|**44.7%**|**0.459**|**68.3%**|**70.9%**|**0.713**|**33.6%**|**37.2%**|**0.323**|
> |DeepSeekVL|PGD|72.8%|77.0%|0.715|0.0%|0.0%|0.037|21.2%|26.4%|0.247|0.0%|0.0%|0.101|
> |DeepSeekVL|CroPA|87.8%|88.6%|0.846|0.0%|0.3%|0.047|57.3%|59.3%|0.521|0.0%|0.0%|0.108|
> |DeepSeekVL|UniAtt|86.3%|88.2%|0.889|1.7%|33.7%|0.265|47.3%|48.9%|0.421|2.2%|27.3%|0.215|
> |DeepSeekVL|**Ours**|**89.1%**|**91.4%**|**0.908**|**44.1%**|**68.9%**|**0.507**|**79.7%**|**80.3%**|**0.798**|**43.2%**|**52.5%**|**0.427**|
>
> **Q2: The paper mentions that it uses all-MiniLM-L6-v2 model to compute the cosine similarity between embeddings of adversarial output and target text. However, it does not explain the reason for using all-MiniLM-L6-v2. Does the cosine similarity computed by other models is consistent with all-MiniLM-L6-v2?**
>
> **A2:** Thanks for your concern. We utilize all-MiniLM-L6-v2 as it is a well-known effective SentenceTransformer model for text embedding. In fact, our attack performance is not sensitive to this embedding encoder. Any latest text encoder can be utilized for the similarity calculation here. Specifically, we implement different text encoders for comparison, including clip-ViT-B-32, all-mpnet-base-v2. As shown in the table below, the results show that our attack can achieve similar performance with different embedding encoders.
>
>
> |Model|Text Encoder|(No Cross) Similarity|(Cross Images/Prompts) Similarity|
> |:----:|:----:|:----:|:----:|
> |LLaVA1.5|all-MiniLM-L6-v2|0.970|0.532|
> |LLaVA1.5|clip-ViT-B-32|0.985|0.541|
> |LLaVA1.5|all-mpnet-base-v2|0.970|0.525|
> |BLIP-2|all-MiniLM-L6-v2|0.970|0.531|
> |BLIP-2|clip-ViT-B-32|0.984|0.563|
> |BLIP-2|all-mpnet-base-v2|0.969|0.530|
> |MiniGPT-4|all-MiniLM-L6-v2|0.977|0.513|
> |MiniGPT-4|clip-ViT-B-32|0.996|0.533|
> |MiniGPT-4|all-mpnet-base-v2|0.977|0.514|
>
> **Q3: The datasets used in experiments are limited. More different types of datasets are needed to further make the experiment results more convincing. Is the proposed attack method still effective on other different datasets and other vision-language tasks?**
>
> **A3:** Thanks for your comment. We conduct more experiments on diverse datasets Flickr30k and NoCaps, where Flickr30k is used for visual commonsense reasoning tasks and NoCaps is used for intent understanding tasks. As shown in the table below, our attack still achieves better performance than baseline attacks on different types of datasets, demonstrating our effectiveness and scalability. We will add these experiments in the revision.
>
> |DataSet & Model|Method|(N) EM-ASR|(N) CM-ASR|(N) Similarity|(I) EM-ASR|(I) CM-ASR|(I) Similarity|(P) EM-ASR|(P) CM-ASR|(P) Similarity|(IP) EM-ASR|(IP) CM-ASR|(IP) Similarity|
> |:----:|:----:|:----:|:----:|:----:|:----:|:----:|:----:|:----:|:----:|:----:|:----:|:----:|:----:|
> |Flickr30k&LLaVA1.5|PGD|95.3%|95.3%|0.956|0.0%|0.0%|0.057|21.4%|22.6%|0.228|0.0%|0.0%|0.051|
> |Flickr30k&LLaVA1.5|CroPA|82.3%|82.3%|0.836|0.2%|0.3%|0.066|70.2%|72.5%|0.727|0.0%|0.0%|0.053|
> |Flickr30k&LLaVA1.5|UniAtt|80.3%|82.9%|0.822|22.1%|23.6%|0.237|66.3%|69.2%|0.708|20.3%|22.0%|0.217|
> |Flickr30k&LLaVA1.5|**Ours**|**98.0%**|**98.0%**|**0.981**|**56.0%**|**56.5%**|**0.589**|**82.1%**|**86.3%**|**0.865**|**27.6%**|**35.2%**|**0.399**|
> |Flickr30k&BLIP-2|PGD|60.9%|62.3%|0.621|0.0%|0.0%|0.076|24.6%|26.5%|0.276|0.0%|0.0%|0.046|
> |Flickr30k&BLIP-2|CroPA|30.4%|92.1%|0.625|0.2%|0.3%|0.116|29.1%|67.6%|0.493|0.0%|0.0%|0.054|
> |Flickr30k&BLIP-2|UniAtt|**76.3%**|79.2%|**0.786**|21.5%|23.7%|0.233|56.1%|58.3%|0.577|19.4%|19.6%|0.205|
> |Flickr30k&BLIP-2|**Ours**|68.4%|**96.1%**|0.709|**52.7%**|**52.7%**|**0.565**|**57.2%**|**84.1%**|**0.723**|**36.7%**|**44.8%**|**0.460**|
> |Flickr30k&MiniGPT-4|PGD|75.6%|75.6%|0.769|0.0%|0.0%|0.067|17.3%|17.6%|0.175|0.0%|0.0%|0.023|
> |Flickr30k&MiniGPT-4|CroPA|94.1%|94.6%|0.948|0.0%|0.0%|0.076|71.6%|72.1%|0.720|0.0%|0.0%|0.038|
> |Flickr30k&MiniGPT-4|UniAtt|82.4%|83.5%|0.843|24.7%|26.5%|0.266|68.2%|69.4%|0.701|19.1%|20.3%|0.199|
> |Flickr30k&MiniGPT-4|**Ours**|**96.1%**|**96.1%**|**0.966**|**34.5%**|**53.6%**|**0.357**|**91.0%**|**91.5%**|**0.932**|**35.6%**|**38.3%**|**0.362**|
> |NoCaps&LLaVA1.5|PGD|95.8%|95.8%|**0.961**|0.0%|0.0%|0.085|37.7%|38.4%|0.396|0.0%|0.0%|0.045|
> |NoCaps&LLaVA1.5|CroPA|76.4%|76.4%|0.781|0.0%|0.0%|0.086|65.3%|66.7%|0.669|0.0%|0.0%|0.036|
> |NoCaps&LLaVA1.5|UniAtt|81.7%|81.7%|0.836|22.3%|24.1%|0.238|69.2%|70.9%|0.713|19.8%|22.1%|0.217|
> |NoCaps&LLaVA1.5|**Ours**|**96.1%**|**96.1%**|0.960|**64.3%**|**65.0%**|**0.694**|**91.6%**|**93.0%**|**0.936**|**42.6%**|**44.3%**|**0.488**|
> |NoCaps&BLIP-2|PGD|71.5%|72.0%|0.723|0.0%|0.0%|0.079|20.3%|20.9%|0.221|0.0%|0.0%|0.41|
> |Flickr30k&BLIP-2|CroPA|29.6%|91.6%|0.630|0.0%|0.0%|0.086|27.5%|71.1%|0.551|0.0%|0.0%|0.075|
> |NoCaps&BLIP-2|UniAtt|**78.1%**|80.6%|0.803|19.6%|21.3%|0.204|56.9%|58.6%|0.573|17.9%|18.3%|0.181|
> |NoCaps&BLIP-2|**Ours**|68.2%|**98.0%**|**0.820**|**55.8%**|**57.4%**|**0.588**|**57.8%**|**81.9%**|**0.684**|**35.5%**|**42.4%**|**0.452**|
> |NoCaps&MiniGPT-4|PGD|92.7%|92.7%|0.935|0.0%|0.0%|0.075|31.6%|32.3%|0.328|0.0%|0.0%|0.064|
> |NoCaps&MiniGPT-4|CroPA|93.1%|93.5%|0.942|0.0%|0.0%|0.063|71.9%|72.6%|0.731|0.0%|0.0%|0.034|
> |NoCaps&MiniGPT-4|UniAtt|81.6%|82.9%|0.820|19.2%|19.7%|0.199|69.2%|70.6%|0.713|17.6%|17.9%|0.183|
> |NoCaps&MiniGPT-4|**Ours**|**98.0%**|**100.0%**|**0.984**|**20.2%**|**59.6%**|**0.366**|**96.1%**|**97.1%**|**0.966**|**26.7%**|**44.7%**|**0.357**|
>
> **Q4: The paper only compare Gaussian distribution and fixed distribution to prove the superior performance of Gaussian distribution. More different types of distributions should be considered and discussed.**
>
> **A4:** Thanks for your comment. We provide more different types of distributions for comparison, including Uniform distribution and Gamma distribution, as shown in the table below. Since the Gaussian distribution is more suitable for large-size sample of great diversity, our modeled distributions through Laplacian approximation achieve the best performance.
>
> |Model|Distribution|(Cross Images) Similarity|(Cross Prompts) Similarity|(Cross Images/Prompts) Similarity|
> |:----:|:----:|:----:|:----:|:----:|
> |LLaVA-1.5|Uniform|0.353|0.927|0.245|
> |LLaVA-1.5|Gamma|0.132|0.828|0.085|
> |LLaVA-1.5|**Gaussian**|**0.688**|**0.945**|**0.644**|
> |BLIP-2|Uniform|0.437|0.562|0.239|
> |BLIP-2|Gamma|0.221|0.427|0.046|
> |BLIP-2|**Gaussian**|**0.582**|**0.671**|**0.522**|
> |MiniGPT-4|Uniform|0.403|0.961|0.314|
> |MiniGPT-4|Gamma|0.297|0.916|0.128|
> |MiniGPT-4|**Gaussian**|**0.661**|**0.973**|**0.355**|
>
> **Q5: I suggest authors to add discussions on limitations in the paper instead of appendix.**
>
> **A5:** Thanks for your suggestion. We will move the limitation discussion section from the appendix to the main paper in the revision.

---

> > ### Comment · Reviewer_RVog · 2025-08-06
> >
> > Thank the authors for answering my questions in the rebuttal. I agree that the method proposed in the paper is clear and mostly effective. I believe that the paper can be improved with the additions. However, the method is still empirical and lacks theoretical analysis. For this reason, I am still leaning toward acceptance rather than rejection, but not a very clear acceptance.  I would like to keep my rating.

---

> > > ### Author Response · Authors · 2025-08-06
> > >
> > > Thank you for your feedback and for acknowledging the clarity and general effectiveness of our method. It is great to know that your questions have been answered.
> > >
> > > Regarding your newly raised concern "the method is still empirical and lacks theoretical analysis", we respectfully clarify that our method is not merely empirical, but is instead founded on solid theoretical principles. We have written detailed theoretical derivations in our paper, with a particular focus on clarifying the rationality of using Laplace approximation as the theoretical basis for Gaussian distribution modeling and integrating Gaussian distribution calculation with the specific MLLM process. Below, we elaborate on our theoretical motivation, mathematical foundation, integration into MLLMs, and empirical verificationaiming to address your concerns regarding the theoretical grounding of our work.
> > >
> > > **Motivation**
> > >
> > > Most existing adversarial MLLM attack methods in multimodal settings rely on instance-specific perturbations, which often lack generalization ability and incur high computational costs. To address this, inspired by Laplace approximation, we model the Gaussian distributions of images and prompts to capture the statistical essence of input data (image and prompt), aiming to generate universal perturbations with strong generalization capabilities and overcome the limitations of existing methods.
> > >
> > > **Mathematical Rigor of Laplace Approximation**
> > >
> > > Our Gaussian distribution modeling is not an empirical assumption but is based on the rigorous mathematical derivation of the Laplace approximation. Laplace approximation, as a classic method for distribution approximation, has been proven by studies such as [19, 20] to efficiently and asymptotically accurately approximate the underlying probability density function, providing a mature theoretical framework for our input distribution modeling. Laplace approximation approximates complex high-dimensional distributions as Gaussian distributions $p(t)=\frac{N^{d/2}}{(2\pi)^{d/2}|{\Sigma}_t|^{1/2}} \exp\left(-\frac{N}{2}(t - \mu_t)^T {\Sigma}_t^{-1}(t - \mu_t)\right)$ (Eq.2) through local quadratic Taylor expansion $\log p(t | v, t_0) \approx \log p(t_v | v, t_0) - \frac{1}{2} (t - t_v)^T H_t (t - t_v)$ (Eq.3) of probability distributions. Its core theoretical basis includes: For a smooth and twice-differentiable probability density function, by performing a second-order Taylor expansion at the mode (the most probable prompt $t_v $ for prompt distribution and the mean value $ \hat{v}$ for image distribution in our scenario), the original distribution of image and prompt can be approximated as a Gaussian distribution with this point as the mean and the inverse of the Hessian matrix as the covariance $t \sim \mathcal{N}(t_v, H_t^{-1})$ (Eq.6) for prompt and $v \sim \mathcal{N}(\hat{v}, H_v^{-1})$ (Eq.7) for image. This process strictly follows the principle of local approximation of probability density functions, ensuring that the calculation of Gaussian distribution parameters (mean, covariance) has a clear mathematical basis. We have introduced more relevant details and justification in the whole Appendix A.
> > >
> > > **Specific to MLLMs**
> > >
> > > To validate the theoretical soundness of embedding our Gaussian distribution modeling into the MLLM workflow, we rigorously integrate Laplace approximation with the model's inherent multimodal processing mechanisms. By minimizing the difference between the MLLMs output and the expected response $L(v, t) = \| \mathrm{MLLM}(v, t) - y _ {\mathrm{exp}} \| _ 2^2$ (Eq.8), we get the $t _ v$ and $H _ t^{-1} = \frac{\|t _ 0 - t _ v\| _ 2^2}{2 \cdot (L(v, t _ 0) - L(v, t _ v))}$ (Eq.10) of prompt distribution, and we calculate the image distribution by getting the mean value of samples $\hat{v} = \Sigma _ {i=1}^N v/{N}$ and directly calculate the covariance $H _ v^{-1} = \frac{1}{N} \Sigma _ v, \quad \text{where} \ \ \Sigma _ v = \frac{1}{N} \sum _ {i=1}^N (v _ i - \hat{v})(v _ i - \hat{v})^\top$ (Eq.11). These calculations are tightly coupled with the MLLM's forward and backward propagation processes: the loss function $L(v, t)$ is computed using the MLLM's output, and the gradients required for optimizing $t _ v$ and estimating Hessian matrices are derived from backpropagating through the MLLM's layers, ensuring the Gaussian distribution parameters are not only theoretically grounded but also specifically tailored to the MLLM's unique processing of multimodal inputs.
> > >
> > > **Verification by Experiments**
> > >
> > > At the same time, we compared the "Gaussian distribution based on Laplace approximation" with some other distributions through the experiments, and the results showed that the former performed better in cross-image/prompt attacks, indirectly verifying the effectiveness of the theoretical support.

---

### Official Review · Reviewer_kztc · 2025-07-01

**Clarity:** 2
**Significance:** 3
**Originality:** 3
**Rating:** 4
**Confidence:** 4

**Summary:**

This paper proposes a novel adversarial attack against MLLMs under the cross-image/prompt scenario. Specifically, it models the potential image-prompt input distribution using Laplace approximation and optimizes a universal adversarial perturbation via Monte Carlo sampling from the distribution. Extensive experiments on prevalent MLLMs across multiple datasets demonstrate the strong generalizability and transferability across unseen image-prompt pairs.

**Questions:**

Please refer to the Weaknesses.

**Ethical Concerns:**

["NO or VERY MINOR ethics concerns only"]

**Final Justification:**

Most comments have been adequately addressed; the paper is now acceptable.

**Limitations:**

Yes.

**Paper Formatting Concerns:**

None.

**Quality:**

3

**Strengths And Weaknesses:**

## Strengths

- **Innovation**: The distribution-driven perspective is innovative, addressing the overfitting issue of existing attacks.
- **Performance**: The experimental results demonstrate that the proposed method outperforms previous attacks.
- **Writing**: The paper is well-structured and easy to follow.

## Weaknesses

- Comparisons with more related adversarial attacks could make it more thorough to validate the superiority.
- The robustness evaluation only contains two defenses. Analyses on more defenses (e.g. defenses against traditional adversarial attacks) are expected.

**Minor Weakness**:
- The placement of Figure 1 and Tables 2, 3, and 5 is inappropriate, affecting the visual flow of the text.
- Table 4 should be positioned after Table 2 and 3.
- The pipeline in Figure 2 is relatively cluttered. It is recommended to redraw it to clarify the connections between different steps.

---

> ### Author Rebuttal · Authors · 2025-07-30
>
> **Q1: Comparisons with more related adversarial attacks could make it more thorough to validate the superiority.**
>
> **A1:** Thanks for your suggestion. We re-implement more related adversarial attacks on No Cross (N) and Cross Images/Prompts (IP) settings to provide a comprehensive experimental comparison, including baselines MFAttack ("On evaluating adversarial robustness of large vision-language models"), Anydoor ("Test-Time Backdoor Attacks on Multimodal Large Language Models"), and XTransfer ("X-Transfer Attacks: Towards Super Transferable Adversarial Attacks on CLIP"). As shown in the table below, our attack still achieves better cross-image/prompt attack performance than these latest attacks, demonstrating our effectiveness. We will add these comparisons in the revision.
>
> |Model|Attack Method|(N) EM-ASR|(N) CM-ASR|(N) Similarity|(IP) EM-ASR|(IP) CM-ASR|(IP) Similarity|
> |:----:|:----:|:----:|:----:|:----:|:----:|:----:|:----:|
> |LLaVA1.5|MFAttack|77.2%|83.0%|0.794|19.7%|21.1%|0.207|
> |LLaVA1.5|Anydoor|71.4%|74.3%|0.734|13.9%|15.3%|0.164|
> |LLaVA1.5|XTransfer|82.6%|87.5%|0.847|27.0%|33.2%|0.325|
> |LLaVA1.5|**Ours**|**96.7%**|**97.3%**|**0.970**|**49.8%**|**51.4%**|**0.532**|
> |BLIP-2|MFAttack|72.4%|76.6%|0.775|13.7%|17.2%|0.179|
> |BLIP-2|Anydoor|73.4%|75.3% |0.749|14.5%|15.1%|0.156|
> |BLIP-2|XTransfer|**79.3%**|82.9%|0.802|18.1%|23.8%|0.203|
> |BLIP-2|**Ours**|49.3%|**98.0%**|**0.970**|**50.0%**|**57.3%**|**0.531**|
> |MiniGPT-4|MFAttack|78.6%|84.2%|0.827|11.0%|22.7%|0.189|
> |MiniGPT-4|Anydoor|72.7%|75.8%|0.768|11.8%|13.3%|0.139|
> |MiniGPT-4|XTransfer|84.8%|88.6%|0.854|28.4%|33.9%|0.310|
> |MiniGPT-4|**Ours**|**97.3%**|**98.0%**|**0.977**|**47.2%**|**57.9%**|**0.513**|
>
> **Q2: The robustness evaluation only contains two defenses. Analyses on more defenses (e.g. defenses against traditional adversarial attacks) are expected.**
>
> **A2:** Thanks for your comment. We implement more defenses, including traditional defense strategies adversarial training, multimodal attention masking, and the latest MLLM defense FARE (ICML2024) of paper "Robust CLIP: Unsupervised Adversarial Fine-Tuning of Vision Embeddings for Robust Large Vision-Language Models", DPS (ICML2025) of paper "Defending LVLMs Against Vision Attacks through Partial-Perception Supervision", to provide more convincing robustness evaluation. We follow the same setting in Figure 5 of our paper for comparison. As shown in the table below, experimental results demonstrate that our proposed cross-image/prompt attack still achieves better transferability under various defenses, demonstrating our effectiveness and robustness. We will provide these defense experiments in the revision.
>
> |Defense|Attack| No Cross (similarity)|Cross Images/Prompts (similarity)|
> |:----:|:----:|:----:|:----:|
> |Adversarial Training|PGD|0.258|0.021|
> |Adversarial Training|CroPA|0.264|0.013|
> |Adversarial Training|UniAtt|0.445|0.161|
> |Adversarial Training|**Ours**|**0.593**|**0.512**|
> |Multimodal Attention Masking|PGD|0.377|0.023|
> |Multimodal Attention Masking|CroPA|0.394|0.036|
> |Multimodal Attention Masking|UniAtt|0.547|0.193|
> |Multimodal Attention Masking|**Ours**|**0.669**|**0.587**|
> |FARE|PGD|0.299|0.047|
> |FARE|CroPA|0.306|0.036|
> |FARE|UniAtt|0.436|0.157|
> |FARE|**Ours**|**0.575**|**0.506**|
> |DPS|PGD|0.274|0.044|
> |DPS|CroPA|0.281|0.036|
> |DPS|UniAtt|0.429|0.146|
> |DPS|**Ours**|**0.567**|**0.512**|
>
> **Q3: Minor Weakness: (1)The placement of Figure 1 and Tables 2, 3, and 5 is inappropriate, affecting the visual flow of the text. (2)Table 4 should be positioned after Tables 2 and 3. (3)The pipeline in Figure 2 is relatively cluttered. It is recommended to redraw it to clarify the connections between different steps.**
>
> **A3:** Thanks for your comments. (1) We will adjust the positions of Figure 1 and Tables 2, 3, and 5 to enhance the visual flow of the text in the revision.  (2) We will move the position of Table 4 after Tables 2 and 3 in the revision. (3) We will redraw and reorganize Figure 2 to display each step more clearly in the revision. We guarantee that all these modifications will be completed in the revision.

---

> > ### Comment · Reviewer_kztc · 2025-08-06
> >
> > Thanks for the authors' effort. Their explanation is reasonable, and I will keep my accept rating.

---

> > > ### Author Response · Authors · 2025-08-06
> > >
> > > Thank you for your valuable response. We appreciate your engagement and will refine the work based on your suggestions.

---

### Official Review · Reviewer_sgCL · 2025-07-03

**Clarity:** 3
**Significance:** 3
**Originality:** 3
**Rating:** 4
**Confidence:** 3

**Summary:**

This paper proposes a novel cross-image/prompt adversarial attack method against multimodal large language models (MLLMs). The key idea is to model the distribution of image and prompt inputs using Laplace approximation, approximating them as Gaussian distributions. The authors then apply Monte Carlo sampling from these distributions to optimize a universal, input-agnostic adversarial perturbation that generalizes across unseen image-prompt pairs. Extensive experiments on LLaVA, BLIP-2, and MiniGPT-4, across MS-COCO and DALLE-3 datasets, show the proposed method achieves superior cross-image/prompt attack performance compared to existing baselines (e.g., PGD, CroPA, UniAtt).

**Questions:**

See weaknesses.

**Ethical Concerns:**

["NO or VERY MINOR ethics concerns only"]

**Limitations:**

1. The method assumes that the modeled input distributions align well with unseen real-world inputs. In practice, distribution shift may degrade attack transferability.
2. The study focuses on simple input-level defenses; it does not explore more advanced defenses (e.g., adversarial training, multimodal attention masking).

**Quality:**

3

**Strengths And Weaknesses:**

## Strengths:
1. Innovative distribution-based attack framework.
2. Strong cross-sample results across multiple datasets and models.
3. Efficiency: generates universal perturbations with reduced computational cost.

## Weaknesses:
1. The entire method and experiments assume full access to model internals, including gradients and architecture. The paper does not explore black-box, gray-box, or query-limited scenarios, which are far more relevant in real-world adversarial contexts where model parameters are typically inaccessible.
2. Experiments only cover MS-COCO and DALLE-3; evaluation on more varied datasets (e.g., Flickr30k) would strengthen generalization claims.

---

> ### Author Rebuttal · Authors · 2025-07-30
>
> **Q1: The entire method and experiments assume full access to model internals, including gradients and architecture. The paper does not explore black-box, gray-box, or query-limited scenarios, which are far more relevant in real-world adversarial contexts where model parameters are typically inaccessible.**
>
> **A1:** Thanks for your concern. We first want to claim that our work proposes to address an important issue of white-box cross-sample transferability, revealing the critical universality of attacks when facing redundant data scenarios in reality. Second, following your suggestion, we can extend our work to the practical scenario without using model gradients and architecture. Since gray-box setting still requires the knowledge of the MLLMs' visual encoder, we implement our attack in the most challenging real-world query-based black-box setting without using any prior knowledge of model parameters.
>
> Specifically, since our distribution modelling and optimization components are independent to the adversarial learning, we can easily replace our white-box gradient backpropagation in Eq.(12) with the general black-box gradient estimation mechanism to estimate the gradient via iterative querying for optimization. As shown in the table below, we evaluate our attack performances on No Cross (N) and Cross Images/Prompts (IP) in the black-box setting with different query numbers. We can find that: (1) Our attack achieves good performance with sufficient query numbers in the black-box setting. (2) Our attack can also achieve competitive performance with limited queries. The above findings indicate that our attack can also handle the real-world black-box cases well, demonstrating the effectiveness and practicality of our proposed method.
>
> |Model|Attack|(N) EM-ASR|(N) CM-ASR|(N) Similarity|(IP) EM-ASR|(IP) CM-ASR|(IP) Similarity|
> |:----:|:----:|:----:|:----:|:----:|:----:|:----:|:----:|
> |LLaVA1.5|Ours (White-box)|95.3%|96.0%|0.958|60.9%|61.3%|0.644|
> |LLaVA1.5|Ours (Black-box, 1000 query)|65.5%|70.1%|0.804|30.2%|32.0%|0.412|
> |LLaVA1.5|Ours (Black-box, 2000 query)|75.8%|80.9%|0.861|    38.4%|42.7%|0.489|
> |LLaVA1.5|Ours (Black-box, 4000 query)|83.6%|88.0%|0.905|47.1%|51.6%|0.566|
> |MiniGPT-4|Ours (White-box)|96.7%|99.3%|0.972|30.8%|52.4%|0.355|
> |MiniGPT-4|Ours (Black-box, 1000 query)|55.4%|69.8%|0.731|12.0%|30.3%|0.217|
> |MiniGPT-4|Ours (Black-box, 2000 query)|70.5%|83.1%|0.827|18.6%|41.7%|0.284|
> |MiniGPT-4|Ours (Black-box, 4000 query)|81.3%|91.4%|0.889|25.0%|48.9%|0.332|
>
> **Q2: Experiments only cover MS-COCO and DALLE-3; evaluation on more varied datasets (e.g., Flickr30k) would strengthen generalization claims.**
>
> **A2:** Thanks for your suggestion. First, we have provided experiments on more dataset SVIT in Appendix B.2. Second, we implement our attacks with more diverse datasets Flickr30k and NoCaps, on No Cross (N), Cross Images (I), Cross Prompts (P) and Cross Images/Prompts (IP) settings. In particular, we conduct visual commonsense reasoning tasks on Flickr30k and intent understanding tasks on NoCaps. As shown in the table below, our attack is still effective on diverse datasets and achieves better cross-image/prompt attack performance compared to other attacks.
>
> |DataSet & Model|Method|(N) EM-ASR|(N) CM-ASR|(N) Similarity|(I) EM-ASR|(I) CM-ASR|(I) Similarity|(P) EM-ASR|(P) CM-ASR|(P) Similarity|(IP) EM-ASR|(IP) CM-ASR|(IP) Similarity|
> |:----:|:----:|:----:|:----:|:----:|:----:|:----:|:----:|:----:|:----:|:----:|:----:|:----:|:----:|
> |Flickr30k&LLaVA1.5|PGD|95.3%|95.3%|0.956|0.0%|0.0%|0.057|21.4%|22.6%|0.228|0.0%|0.0%|0.051|
> |Flickr30k&LLaVA1.5|CroPA|82.3%|82.3%|0.836|0.2%|0.3%|0.066|70.2%|72.5%|0.727|0.0%|0.0%|0.053|
> |Flickr30k&LLaVA1.5|UniAtt|80.3%|82.9%|0.822|22.1%|23.6%|0.237|66.3%|69.2%|0.708|20.3%|22.0%|0.217|
> |Flickr30k&LLaVA1.5|**Ours**|**98.0%**|**98.0%**|**0.981**|**56.0%**|**56.5%**|**0.589**|**82.1%**|**86.3%**|**0.865**|**27.6%**|**35.2%**|**0.399**|
> |Flickr30k&BLIP-2|PGD|60.9%|62.3%|0.621|0.0%|0.0%|0.076|24.6%|26.5%|0.276|0.0%|0.0%|0.046|
> |Flickr30k&BLIP-2|CroPA|30.4%|92.1%|0.625|0.2%|0.3%|0.116|29.1%|67.6%|0.493|0.0%|0.0%|0.054|
> |Flickr30k&BLIP-2|UniAtt|**76.3%**|79.2%|**0.786**|21.5%|23.7%|0.233|56.1%|58.3%|0.577|19.4%|19.6%|0.205|
> |Flickr30k&BLIP-2|**Ours**|68.4%|**96.1%**|0.709|**52.7%**|**52.7%**|**0.565**|**57.2%**|**84.1%**|**0.723**|**36.7%**|**44.8%**|**0.460**|
> |Flickr30k&MiniGPT-4|PGD|75.6%|75.6%|0.769|0.0%|0.0%|0.067|17.3%|17.6%|0.175|0.0%|0.0%|0.023|
> |Flickr30k&MiniGPT-4|CroPA|94.1%|94.6%|0.948|0.0%|0.0%|0.076|71.6%|72.1%|0.720|0.0%|0.0%|0.038|
> |Flickr30k&MiniGPT-4|UniAtt|82.4%|83.5%|0.843|24.7%|26.5%|0.266|68.2%|69.4%|0.701|19.1%|20.3%|0.199|
> |Flickr30k&MiniGPT-4|**Ours**|**96.1%**|**96.1%**|**0.966**|**34.5%**|**53.6%**|**0.357**|**91.0%**|**91.5%**|**0.932**|**35.6%**|**38.3%**|**0.362**|
> |NoCaps&LLaVA1.5|PGD|95.8%|95.8%|**0.961**|0.0%|0.0%|0.085|37.7%|38.4%|0.396|0.0%|0.0%|0.045|
> |NoCaps&LLaVA1.5|CroPA|76.4%|76.4%|0.781|0.0%|0.0%|0.086|65.3%|66.7%|0.669|0.0%|0.0%|0.036|
> |NoCaps&LLaVA1.5|UniAtt|81.7%|81.7%|0.836|22.3%|24.1%|0.238|69.2%|70.9%|0.713|19.8%|22.1%|0.217|
> |NoCaps&LLaVA1.5|**Ours**|**96.1%**|**96.1%**|0.960|**64.3%**|**65.0%**|**0.694**|**91.6%**|**93.0%**|**0.936**|**42.6%**|**44.3%**|**0.488**|
> |NoCaps&BLIP-2|PGD|71.5%|72.0%|0.723|0.0%|0.0%|0.079|20.3%|20.9%|0.221|0.0%|0.0%|0.41|
> |Flickr30k&BLIP-2|CroPA|29.6%|91.6%|0.630|0.0%|0.0%|0.086|27.5%|71.1%|0.551|0.0%|0.0%|0.075|
> |NoCaps&BLIP-2|UniAtt|**78.1%**|80.6%|0.803|19.6%|21.3%|0.204|56.9%|58.6%|0.573|17.9%|18.3%|0.181|
> |NoCaps&BLIP-2|**Ours**|68.2%|**98.0%**|**0.820**|**55.8%**|**57.4%**|**0.588**|**57.8%**|**81.9%**|**0.684**|**35.5%**|**42.4%**|**0.452**|
> |NoCaps&MiniGPT-4|PGD|92.7%|92.7%|0.935|0.0%|0.0%|0.075|31.6%|32.3%|0.328|0.0%|0.0%|0.064|
> |NoCaps&MiniGPT-4|CroPA|93.1%|93.5%|0.942|0.0%|0.0%|0.063|71.9%|72.6%|0.731|0.0%|0.0%|0.034|
> |NoCaps&MiniGPT-4|UniAtt|81.6%|82.9%|0.820|19.2%|19.7%|0.199|69.2%|70.6%|0.713|17.6%|17.9%|0.183|
> |NoCaps&MiniGPT-4|**Ours**|**98.0%**|**100.0%**|**0.984**|**20.2%**|**59.6%**|**0.366**|**96.1%**|**97.1%**|**0.966**|**26.7%**|**44.7%**|**0.357**|
>
> **Q3: The method assumes that the modeled input distributions align well with unseen real-world inputs. In practice, distribution shift may degrade attack transferability.**
>
> **A3:** Thanks for your concern. In fact, since we model the image distribution with sufficient natural images and model the text distribution with comprehensive task-relevant/-irrelevant prompts, our approximated Gaussian distribution is generalizable enough to handle the general image-prompt inputs in real-world cases. As proved in Table 8 and Table 9 of the appendix, our method can still achieve effective transfer-attack performance when transferring our generated perturbations across distribution-different datasets. The results shown in the table below also demonstrate that our attack is robust to distribution shift and is scalable to be further improved with more diverse data usage.
>
> |Transfer|Model|EM-ASR|CM-ASR|Similarity|
> |:----:|:----:|:----:|:----:|:----:|
> |MS-COCO to MS-COCO|LLaVA-1.5|49.8%|51.4%|0.532|
> |MS-COCO to MS-COCO|BLIP-2|50.0%|57.3%|0.531|
> |MS-COCO to MS-COCO|MiniGPT-4|47.2%|57.9%|0.513|
> |MS-COCO to DALLE-3|LLaVA-1.5|49.3%|50.3%|0.526|
> |MS-COCO to DALLE-3|BLIP-2|29.8%|40.1%|0.363|
> |MS-COCO to DALLE-3|MiniGPT-4|26.8%|34.0%|0.307|
> |MS-COCO+SVIT to DALLE-3|LLaVA-1.5|51.7%|53.6%|0.557|
> |MS-COCO+SVIT to DALLE-3|BLIP-2|35.6%|48.1%|0.435|
> |MS-COCO+SVIT to DALLE-3|MiniGPT-4|34.5%|40.8%|0.329|
>
> **Q4: The study focuses on simple input-level defenses; it does not explore more advanced defenses (e.g., adversarial training, multimodal attention masking).**
>
> **A4:** Thanks for your comment. We implement more advanced defenses to provide convincing robustness evaluation, including adversarial training, multimodal attention masking, and MLLM defense FARE (ICML2024) of paper "Robust CLIP: Unsupervised Adversarial Fine-Tuning of Vision Embeddings for Robust Large Vision-Language Models", DPS (ICML2025) of paper "Defending LVLMs Against Vision Attacks through Partial-Perception Supervision". We follow the same setting in Figure 5 of our paper for comparison. As shown in the table below, the experimental results demonstrate that our proposed attack still achieves better transferability across images and prompts under various defenses, demonstrating our effectiveness and robustness. We will provide these defense experiments in the revision.
>
> |Defense|Attack| No Cross (similarity)|Cross Images/Prompts (similarity)|
> |:----:|:----:|:----:|:----:|
> |Adversarial Training|PGD|0.258|0.021|
> |Adversarial Training|CroPA|0.264|0.013|
> |Adversarial Training|UniAtt|0.445|0.161|
> |Adversarial Training|**Ours**|**0.593**|**0.512**|
> |Multimodal Attention Masking|PGD|0.377|0.023|
> |Multimodal Attention Masking|CroPA|0.394|0.036|
> |Multimodal Attention Masking|UniAtt|0.547|0.193|
> |Multimodal Attention Masking|**Ours**|**0.669**|**0.587**|
> |FARE|PGD|0.299|0.047|
> |FARE|CroPA|0.306|0.036|
> |FARE|UniAtt|0.436|0.157|
> |FARE|**Ours**|**0.575**|**0.506**|
> |DPS|PGD|0.274|0.044|
> |DPS|CroPA|0.281|0.036|
> |DPS|UniAtt|0.429|0.146|
> |DPS|**Ours**|**0.567**|**0.512**|

---

### Official Review · Reviewer_8ePi · 2025-07-21

**Clarity:** 3
**Significance:** 4
**Originality:** 3
**Rating:** 5
**Confidence:** 4

**Summary:**

This paper introduces a new approach to generate cross-image/prompt transferable adversarial examples by approximating the distribution of both image and text input. This method achieves a significant improvement compared to the previous methods.

**Questions:**

1. I notice the optimization step number is set to 300, while CroPA uses 2000. Will a larger step number lead to better performance?
2. The author can also show the cross-model transferability of the proposed method, and compare the result with methods like [1].
3. Can you provide more details on the efficiency analysis? It generates **all** adversarial examples. But the proposed method generates the cross-image perturbation for all data but CroPA generates adversarial examples for each data sample. Do I understand correctly? And what’s the size of the dataset used?

[1] Gu, Chenhe, et al. "Improving Adversarial Transferability in MLLMs via Dynamic Vision-Language Alignment Attack." arXiv preprint arXiv:2502.19672 (2025).

**Ethical Concerns:**

["NO or VERY MINOR ethics concerns only"]

**Final Justification:**

I believe this is a good paper on the transferable adversarial attack on MLLMs and should be accepted.

**Limitations:**

1. This approach approximates the distribution of a given prompt/image set, which might be ineffective under a prompt or an image from a different distribution. So it is crucial to sample a good prompt/image set.

**Quality:**

3

**Strengths And Weaknesses:**

## Strength

1. The proposed method achieves a significant improvement compared to the previous methods.
2. Based on the simple intuition of approximating the distribution to improve the generalization/transferability of adversarial attacks.

## Weakness

1. Lack of results on state-of-the-art MLLMs, like Qwen2-VL, InternVL, as well as closed-source models.
2. This method introduces a new computation to fit both the image and text distribution, as well as a sampling computation.

---

> ### Author Rebuttal · Authors · 2025-07-30
>
> **Q1: Lack of results on state-of-the-art MLLMs, like Qwen2-VL, InternVL, as well as closed-source models.**
>
> **A1:** Thanks for your suggestion. We provide experimental results on more state-of-the-art MLLMs, Qwen2-VL, InternVL, DeepSeekVL, on No Cross (N), Cross Images (I), Cross Prompts (P), and Cross Images/Prompts (IP) settings, as shown in the table below. It demonstrates that our attack can also achieve better performance against these MLLMs compared to other attacks.
>
> |Model|Method|(N) EM-ASR|(N) CM-ASR|(N) Similarity|(I) EM-ASR|(I) CM-ASR|(I) Similarity|(P) EM-ASR|(P) CM-ASR|(P) Similarity|(IP) EM-ASR|(IP) CM-ASR|(IP) Similarity|
> |:----:|:----:|:----:|:----:|:----:|:----:|:----:|:----:|:----:|:----:|:----:|:----:|:----:|:----:|
> |Qwen2-VL|PGD|63.3%|69.1%|0.676|0.0%|0.0%|0.046|28.4%|31.5%|0.334|0.0%|0.0%|0.059|
> |Qwen2-VL|CroPA|73.8%|76.6%|0.771|0.0%|0.2%|0.044|65.9%|67.5%|0.675|0.0%|0.0%|0.076|
> |Qwen2-VL|UniAtt|76.2%|79.7%|0.792|23.1%|25.6%|0.265|68.3%|69.6%|0.699|20.1%|21.5%|0.231|
> |Qwen2-VL|**Ours**|**81.2%**|**85.2%**|**0.838**|**44.6%**|**60.6%**|**0.487**|**77.2%**|**78.6%**|**0.775**|**42.7%**|**43.6%**|**0.457**|
> |InternVL|PGD|61.0%|67.4%|0.661|0.0%|0.0%|0.075|25.3%|26.8%|0.271|0.0%|0.0%|0.047|
> |InternVL|CroPA|73.1%|75.4%|0.742|0.0%|0.0%|0.086|64.6%|66.2%|0.668|0.0%|0.0%|0.081|
> |InternVL|UniAtt|75.4%|78.2%|**0.779**|22.7%|24.6%|0.248|64.8%|67.1%|0.692|21.6%|23.7%|0.245|
> |InternVL|**Ours**|**75.8%**|**79.0%**|0.745|**43.2%**|**44.7%**|**0.459**|**68.3%**|**70.9%**|**0.713**|**33.6%**|**37.2%**|**0.323**|
> |DeepSeekVL|PGD|72.8%|77.0%|0.715|0.0%|0.0%|0.037|21.2%|26.4%|0.247|0.0%|0.0%|0.101|
> |DeepSeekVL|CroPA|87.8%|88.6%|0.846|0.0%|0.3%|0.047|57.3%|59.3%|0.521|0.0%|0.0%|0.108|
> |DeepSeekVL|UniAtt|86.3%|88.2%|0.889|1.7%|33.7%|0.265|47.3%|48.9%|0.421|2.2%|27.3%|0.215|
> |DeepSeekVL|**Ours**|**89.1%**|**91.4%**|**0.908**|**44.1%**|**68.9%**|**0.507**|**79.7%**|**80.3%**|**0.798**|**43.2%**|**52.5%**|**0.427**|
>
> Since existing methods fail to attack closed-sourced MLLMs while these MLLMs are somewhat expensive, we try to re-implement our attack by replacing our white-box gradient backpropagation in Eq.(12) with the general black-box gradient estimation mechanism to attack closed-sourced MLLMs GPT-4o and Gemini-2.0 in the table below, where our attack is still effective.
>
> |Model|Method|(N) EM-ASR|(N) CM-ASR|(N) Similarity|(IP) EM-ASR|(IP) CM-ASR|(IP) Similarity|
> |:----:|:----:|:----:|:----:|:----:|:----:|:----:|:----:|
> |GPT-4o|Ours|36.8%|52.6%|0.597|25.9%|39.1%|0.304|
> |Gemini-2.0|Ours|38.3%|50.4%|0.562|22.8%|38.5%|0.316|
>
> We will add these experiments in the revision.
>
> **Q2: This method introduces a new computation to fit both the image and text distribution, as well as a sampling computation.**
>
> **A2:** Thanks for your concerns. Although our method introduces new distribution modeling and sampling computations, our attack is still competitively efficient compared to existing works as shown in the table below. (1) These two components are lightweight and their costs can be ignored compared with the overall costs, as the main cost of our attack is the latter gradient optimization process. (2) Our proposed attack is able to generate the universal adversarial example across images/prompts for the whole dataset in a single process, while existing attacks need to generate individual perturbation for each sample of the dataset.
>
> |Method|Component|Runtime|GPU Memory|
> |:----:|:----:|:----:|:----:|
> |Cropa|-|16.2h|23.9GB|
> |UniAtt|-|4.9h|57.5GB|
> |Ours|Modeling Distribution|0.8h|26.5GB|
> |Ours|Sampling|0.1h|2.8GB|
> |Ours|whole process|5.5h|48.3GB|
>
> **Q3: I notice the optimization step number is set to 300, while CroPA uses 2000. Will a larger step number lead to better performance?**
>
> **A3:** As shown in the table below, a larger step number will introduce slightly better performance of our proposed attack. However, to take the balance between the complexity and performance, we set the step number to 300 as it is enough to achieve good performance.
>
> |Epoch Number|(N) EM-ASR|(N) CM-ASR|(N) Similarity|(IP) EM-ASR|(IP) CM-ASR|(IP) Similarity|
> |:----:|:----:|:----:|:----:|:----:|:----:|:----:|
> |300|49.3%|98.0%|0.970|50.0%|57.3%|0.531|
> |2000|50.2%|98.1%|0.969|50.7%|57.9%|0.529|
>
> **Q4: The author can also show the cross-model transferability of the proposed method, and compare the result with methods like [1].**
>
> **A4:** Thanks for your comment. We have provided our cross-model transfer-attack experiments in Table 10 of the Appendix. To provide a detailed comparison, we re-implement your mentioned paper [1] (code is not released) in the table below. Results show that our attack achieves competitive cross-model attack performance, even our method is not specifically designed for cross-model attack.
>
> |Transfer|Method|EM-ASR|CM-ASR|Similarity|
> |:----:|:----:|:----:|:----:|:----:|
> |BLIP-2 to MiniGPT-4|[1]|**10.3%** (re-implemented)|**31.0%** (reported)|0.122 (re-implemented)|
> |BLIP-2 to MiniGPT-4|Ours|0.0%|19.4%|**0.129**|
> |MiniGPT-4 to BLIP-2|[1]|14.4% (reported)|18.1% (re-implemented)|0.127 (re-implemented)|
> |MiniGPT-4 to BLIP-2|Ours|**28.9%**|**32.3%**|**0.351**|
>
>
> **Q5: Can you provide more details on the efficiency analysis? It generates all adversarial examples. But the proposed method generates the cross-image perturbation for all data but CroPA generates adversarial examples for each data sample. Do I understand correctly? And what’s the size of the dataset used?**
>
> **A5:** Your understanding is correct. In Table 3 of the paper, we evaluate the costs of generating 200 adversarial examples from the dataset. For previous work CroPA, it generates adversarial examples for each sample individually, requiring redundant costs of 200 samples. Instead, our method only generates a universal adversarial perturbation in one single process, and this perturbation can be directly added to all samples for attacks without additional costs. Therefore, under the same realistic whole-dataset attack setting, our method achieves better efficiency. We will provide the detailed analysis in the revision.
>
> **Q6: This approach approximates the distribution of a given prompt/image set, which might be ineffective under a prompt or an image from a different distribution. So it is crucial to sample a good prompt/image set.**
>
> **A6:** This is a good question. Indeed, our attack performance is correlated to the sampling image/prompt set. Intuitively, a larger number of diverse samples can guarantee the generalization-ability of the modeled distribution, providing more reliable guidance similar to the real-world cases. Although we do not sample a large amount of data for distribution modeling due to resource costs, our work is proven to be robust enough to potential domain shift by the cross-dataset attack performance in Table 8 and Table 9 of the appendix. This is because we model the image distribution with sufficient natural images and model the text distribution with comprehensive task-relevant/-irrelevant prompts, leading to learn general distributions. The table below further indicates the scalability of our proposed attack where more diverse data helps the distribution modelling for improving the attack performance.
>
> |Transfer|Model|EM-ASR|CM-ASR|Similarity|
> |:----:|:----:|:----:|:----:|:----:|
> |MS-COCO to MS-COCO|LLaVA-1.5|49.8%|51.4%|0.532|
> |MS-COCO to MS-COCO|BLIP-2|50.0%|57.3%|0.531|
> |MS-COCO to MS-COCO|MiniGPT-4|47.2%|57.9%|0.513|
> |MS-COCO to DALLE-3|LLaVA-1.5|49.3%|50.3%|0.526|
> |MS-COCO to DALLE-3|BLIP-2|29.8%|40.1%|0.363|
> |MS-COCO to DALLE-3|MiniGPT-4|26.8%|34.0%|0.307|
> |MS-COCO+SVIT to DALLE-3|LLaVA-1.5|51.7%|53.6%|0.557|
> |MS-COCO+SVIT to DALLE-3|BLIP-2|35.6%|48.1%|0.435|
> |MS-COCO+SVIT to DALLE-3|MiniGPT-4|34.5%|40.8%|0.329|

---

> > ### Comment · Reviewer_8ePi · 2025-08-08
> > **Response**
> >
> > Thanks for the responses and all my concerns are addressed. I will adjust my rating.

---

> > > ### Author Response · Authors · 2025-08-08
> > >
> > > We would like to thank the reviewer for responding to our rebuttal. It is great to know that your concerns have been addressed. Thank you once again for your valuable contributions.

---

> > > ### Author Response · Authors · 2025-08-09
> > >
> > > Dear Reviewer 8ePi,
> > >
> > > Thank you again for your feedback earlier. As the discussion deadline is approaching, we would like to kindly remind you about the possible score adjustment you mentioned.
> > >
> > > Sincerely,
> > >
> > > The Authors of Submission 4817

---

### Decision · Program_Chairs · 2025-09-17

**Decision:**

Accept (poster)

**Comment:**

This paper proposes a novel adversarial attack on Multimodal Large Language Models (MLLMs) that enhances transferability and universality across diverse image–prompt pairs. Unlike prior methods that overfit to specific inputs or datasets, the authors leverage distribution approximation theory—using a Laplace approximation to estimate mean and covariance of multimodal inputs—and Monte Carlo sampling to optimize a single, input-agnostic perturbation. This results in adversarial perturbations with strong generalization, enabling effective cross-image/prompt transfer attacks at lower resource cost. Extensive experiments demonstrate that the proposed method achieves robust adversarial capabilities against a range of state-of-the-art MLLMs. During the rebuttal, the authors successfully addressed all reviewer concerns, and we have decided to accept this work. We encourage the authors to incorporate the promised changes and refinements in the camera-ready version.